# Statistical Analysis of the Consistency of HRV Analysis Using BCG or Pulse Wave Signals

**DOI:** 10.3390/s22062423

**Published:** 2022-03-21

**Authors:** Huiying Cui, Zhongyi Wang, Bin Yu, Fangfang Jiang, Ning Geng, Yongchun Li, Lisheng Xu, Dingchang Zheng, Biyong Zhang, Peilin Lu, Stephen E. Greenwald

**Affiliations:** 1College of Medicine and Biological and Information Engineering, Northeastern University, Shenyang 110167, China; 1971039@stu.neu.edu.cn (H.C.); wangzyppopp@163.com (Z.W.); jiangff@bmie.neu.edu.cn (F.J.); 2Philips Design, 5611 AZ Eindhoven, The Netherlands; bin.yu@philips.com; 3Department of Cardiology, Shengjing Hospital of China Medical University, Shenyang 110819, China; ninggeng@aliyun.com; 4Shenyang Contain Electronic Technology Co., Ltd., Shenyang 110167, China; liyongchun@contain.com.cn; 5Neusoft Research of Intelligent Healthcare Technology, Co., Ltd., Shenyang 110167, China; 6Research Centre for Intelligent Healthcare, Coventry University, Coventry CV1 5RW, UK; ad4291@coventry.ac.uk; 7BOBO Technology, Hangzhou 310000, China; b.zhang@tue.nl; 8User System Interaction Group, Industrial Design, Eindhoven University of Technology, 5612 AZ Eindhoven, The Netherlands; 9Neuroscience Center, Department of Neurology, Sir Run Run Shaw Hospital, School of Medicine, Zhejiang University, Hangzhou 310000, China; zxa2004@zju.edu.cn; 10Blizard Institute, Barts & The London School of Medicine & Dentistry, Queen Mary University of London, London E1 4NS, UK

**Keywords:** heartrate variability, ballistocardiography, pulse wave, electrocardiography, hrv analysis

## Abstract

Ballistocardiography (BCG) is considered a good alternative to HRV analysis with its non-contact and unobtrusive acquisition characteristics. However, consensus about its validity has not yet been established. In this study, 50 healthy subjects (26.2 ± 5.5 years old, 22 females, 28 males) were invited. Comprehensive statistical analysis, including Coefficients of Variation (CV), Lin’s Concordance Correlation Coefficient (LCCC), and Bland-Altman analysis (BA ratio), were utilized to analyze the consistency of BCG and ECG signals in HRV analysis. If the methods gave different answers, the worst case was taken as the result. Measures of consistency such as Mean, SDNN, LF gave good agreement (the absolute value of CV difference < 2%, LCCC > 0.99, BA ratio < 0.1) between J-J (BCG) and R-R intervals (ECG). pNN50 showed moderate agreement (the absolute value of CV difference < 5%, LCCC > 0.95, BA ratio < 0.2), while RMSSD, HF, LF/HF indicated poor agreement (the absolute value of CV difference ≥ 5% or LCCC ≤ 0.95 or BA ratio ≥ 0.2). Additionally, the R-R intervals were compared with P-P intervals extracted from the pulse wave (PW). Except for pNN50, which exhibited poor agreement in this comparison, the performances of the HRV indices estimated from the PW and the BCG signals were similar.

## 1. Introduction

Cardiovascular disease is now the principal threat to human health worldwide [1] and has become a serious challenge to the medical systems of many countries. In recent years, more young people are facing this threat [2]. Unfortunately, many cardiovascular diseases do not give rise to obvious symptoms in their early stages, often leading to delayed diagnosis. However, certain physiological variables, such as heart rate and heart rate variability (HRV), exhibit early changes before other symptoms are seen [3]. Therefore, the monitoring of heart rate and its short-term variability in daily life can be of clinical value in detecting cardiovascular disease at a stage when early treatment is beneficial.

HRV refers to the minor difference between beat-to-beat heart intervals. It may reflect the activity of the sympathetic nervous system (SNS) and parasympathetic nervous system (PNS) in the autonomic nervous system (ANS) [4]. The regular activity of the healthy heart is maintained by the interaction between the sympathetic and parasympathetic nervous systems, but an imbalance in this interaction can lead to cardiovascular dysfunction. It has been known for at least 50 years that changes in HRV are strongly associated with cardiovascular diseases, such as heart failure, coronary artery disease, acute myocardial infarction and essential hypertension, and more recent studies have confirmed this [5,6,7,8]. Therefore, HRV analysis has become important in the screening of human cardiovascular function.

The gold standard for assessing HRV is the analysis of the R-R interval of the ECG signal, followed by the calculation of various descriptive parameters. In clinical practice, the 24-h ECG is generally analyzed because it can accurately and comprehensively reflect the patient’s cardiac activity under a range of conditions [1]. On the other hand, most studies focusing on HRV, rather than the characteristics of the ECG signal itself, have used data collected in the short-term (5-min) [9,10,11,12,13].

The Pulse Wave (PW), detected, for instance, by photoplethysmography or tonometry, is often used for the noninvasive assessment of the cardiovascular system but can also be used as an alternative source to the ECG to record and analyze HRV. Strictly speaking, since pulse signals are used, the results should be referred to as pulse rate variability (PRV). Compared with ECG, the equipment used to acquire the PW signal is simpler and more convenient to use, and is, therefore, more suitable for routine monitoring, providing a good alternative to ECG-based HRV analysis [14,15].

Ballistocardiography (BCG) is a technique that records the force caused by the ejection of blood by the heart into the arteries [16]. The force signal can be acquired noninvasively using polyvinylidene fluoride (PVDF) or ElectroMechanical Film (EMFi) sensors. By recording the force and work of the heartbeats, as well as their timing, BCG signals can also be used to evaluate hemodynamic changes in the cardiovascular system. In recent years, a variety of BCG signal acquisition systems for health monitoring have been designed and developed, most of which come in the form of a mattress, chair, or weighing scale [9,16,17]. Unlike ECG, the BCG signal can be acquired by non-contact sensors with less discomfort and inconvenience. Ease of operation is also an important feature of BCG-monitoring equipment. Overall, the recording of BCG signals is relatively simple and unobtrusive, so that the user is essentially unaware of its presence.

BCG signals have been considered an alternative method for HRV analysis in various studies [2,10,11,12,13]. Alba et al. [10] explored the feasibility of HRV analysis using the J-J, I-I, K-K, and H-H intervals of BCG signals. The results showed that the J-J intervals were largely consistent with simultaneously recorded ECG signals. Jan et al. [11] compared BCG and ECG signals to monitor heart rate. However, only the beat-to-beat intervals were analyzed, without further analysis of any HRV parameters. Brueser et al. [12] discussed the HRV parameters calculated from BCG signals; they chose a set such as pNN50, SDNN, RMSSD, LF, HF and LF/HF. Jae et al. [13] assessed virtually all the widely used HRV parameters, including AVNN, SDNN, RMSSD, pNN50, LF, HF, SD1, SD2, etc. However, they only analyzed the correlation between short-term BCG and ECG signals and the number of subjects was small.

In many studies, there is no consensus regarding the validity of using BCG instead of ECG for HRV analysis. Ville et al. reported that all the time-domain and frequency-domain HRV parameters calculated from BCG and ECG agreed well [18]. However, Christoph et al. found that only for the parameter LF/HF did the BCG and ECG signals show a good correlation [12], whereas Cao et al. found significant differences between pNN50, LF/HF and SD1 [17]. A likely explanation for these different findings is that they used different statistical methods to analyze their results. Most studies used the Pearson correlation and the relative error [10,11,12,13,19]. However, both are measures of correlation rather than agreement between variables, and they are not, therefore, an appropriate means of assessing agreement. While it is true that the results from tests that agree well will have a high Pearson correlation and low relative error, the converse is not always true [20,21]. 

Studies have shown that the PRV can be often used as an alternative for HRV analysis in healthy subjects [19]. However, a wearable sensor is required to record the pulse signals [22], whereas this is not needed to record the BCG signal. We have previously investigated the use of the pulse wave to replace the ECG for HRV analysis and found that the degree of concordance between the two methods varies for different groups of people. In healthy adolescents, the two signals can replace each other for HRV analysis. However, in elderly subjects, and patients with cardiovascular disease, some parameters (pNN50, RMSSD, LF, HF) are not consistent and cannot be substituted for each other [23].

To comprehensively analyze the potential concordance or disagreement between results based on BCG and Pulse wave (PW) signals, we adopted a wider range of statistical methods, including the coefficient of variation, Bland-Altman analysis, and Lin’s concordance correlation coefficient.

The main contributions of this article are as follows:(a)A comprehensive statistical analysis, as outlined in the preceding paragraph.(b)A comparison of the pros and cons of BCG and PW signals as a substitute for ECG when analyzing HRV in young adults.

The remainder of the paper is organized as follows. Section 2 presents the design of the experiment, data collection equipment and the data analysis methods. Section 3 describes and demonstrates the experimental results. In Section 4, we analyze the consistency of the HRV analysis based on BCG signals and compare the pros and cons of BCG signals and PW signals for HRV calculation. Finally, we summarize our findings and outline our aims for future work.

## 2. Materials and Methods

A total of 50 healthy young adult volunteers (age 26.2 ± 5.5 years, 22 females and 28 males) were recruited. The exclusion criteria included the use of medication and any medical condition associated with abnormal heart rate signals. This study has been approved by the research ethics committee of Northeastern University. Study number: NEU-EC-20208015S.

The experiment was divided into two parts. In the first, we collected short-term data (5 min) consisting of ECG, BCG, and PW signals from all 50 subjects to assess the feasibility of calculating HRV parameters based on BCG and PW signals. In the second part, we collected data over a longer time interval (100 min) from three subjects only, to explore the stability of BCG signals in calculating medium-term HRV parameters. The aim was to verify whether BCG and PW signals can be used for medium- to longer-term HRV analysis.

A device was developed by BOBO Technology (Hangzhou, China) to acquire the BCG signal. It consisted of a poly-vinylidene fluoride (PVDF), piezoelectric film sensor (33 cm × 30 cm), integrated into an ESP32 board, combined with a 12-bit SAR ADC and TP1562AL1 rail-to-rail CMOS operational amplifier, set in a smart cushion (type LS-ADA). The device for the synchronous recording of ECG and PW signals was designed in our laboratory; the ECG was detected with Ag/AgCl electrodes on the chest and ankle. A piezoresistive strain gauge sensor (PESG) was used as a tonometer to convert pressure signals into a change in strain-dependent resistance. This was converted, in a bridge circuit, to a varying voltage, which corresponded to the pulse wave signal [24,25,26].

All three signals were recorded at a sampling rate of 1 kHz: the bandwidth of the ECG signal ranged from 0.05 Hz to 100 Hz [27], that of the PW signals was 0.2 Hz to 40 Hz [28], and for the BCG signals, it was 4 Hz to 10 Hz [29]. The two acquisition devices are shown in Figure 1: laptop A was used to record the BCG signals, and laptop B was used to record the PW and ECG signals simultaneously. The synchronization of recordings from the two computers is described below.

Long-term recordings used to monitor HRV are generally continued for 24 h to account for the effect of circadian rhythm. For practical purposes, we designed a BCG signal acquisition device in the form of a seat, which can be used for daily health monitoring. Sitting for 24 h is obviously unreasonable. Therefore, in order to compare the HRV obtained by the three modalities over a sufficient time period to allow for medium-term variation in cardiovascular control mechanisms, we chose 100 min, and defined this as a “medium-term” data acquisition period.

At the start of the recording session, subjects were asked to relax for 1 min before data collection to reduce the effect of body movement on the BCG signal. In the process of collecting data, the subjects were asked to keep as still as possible. For each subject, the signal was recorded for a total of 100 min, but this period was divided into 20 5-min intervals, each followed by a 2-min rest period. Although this procedure helped to reduce movement artefacts, they were not entirely eliminated, and the signals required further processing (see below for more details).

As mentioned above and shown in Figure 1, we used two independent signal acquisition devices, so precise synchronization of all the data was not possible in real time. Synchronization was realized off-line before further signal processing by finding the maximum of the cross correlation between the BCG and ECG signals. The PW signals were band-pass filtered (cut-off frequencies 0.5 Hz and 10 Hz), the ECG signals were passed through a 50 Hz notch filter, and the noise in the BCG signal was removed with a 5th-order ‘db4’ wavelet filter using its default threshold. All processing was performed in MATLAB R2018a).

Studies have shown that there are slight differences in the measured beat-to-beat intervals obtained from different types of recording devices. However, the R-R interval, as a measure of heart interval, has become the gold standard in current research [30]. In this study, we chose R-R, P-P, J-J intervals to calculate the heartrate from the ECG, the PW and BCG, respectively. The J wave of the BCG and the first peak of PW signals were taken as the maximum amplitude points in each period. For the ECG, we detected the R wave by first finding the maximum value of the first derivative and then using this as a marker to find the maximum value of the points in the undifferentiated signal immediately following this. Figure 2 shows the peaks detected from a typical set of signals. Intervals less than 350 ms or greater than 1500 ms, or those differing by more than 20% of the value of the preceding and subsequent interval, were treated as artefacts, defined as spurious and replaced by a linear interpolation of the values derived from the neighboring peaks [7], Table 1 lists the number of spurious peaks detected by each signal.

HRV analysis includes time domain methods, frequency domain methods and nonlinear analysis. In this paper, we chose the time domain parameters mean, SDNN, pNN50 and RMSSD; the frequency domain parameters LF, HF and LF/HF; and the non-linear analysis parameters SD1 and SD2, derived from Poincaré plots. However, SD1 and RMSSD are essentially equivalent, as are SD2 and SDNN [31,32]. Therefore, only RMSSD and SDNN were analyzed here, in place of SD1 and SD2. On the other hand, the results of nonlinear analysis are affected by many factors, such as the method used to preprocess the data, parameter settings, inconsistent algorithms for calculating the parameters, individual differences between subjects, etc. For example, entropy is a commonly used index in HRV nonlinear analysis and in all entropy measurement methods, the embedding dimension and delay time play a decisive role as initialization parameters. However, there is no standard for the optimal value of initialization parameters, which are mainly determined according to a priori knowledge and data structure. This means that, under different initialization parameters, the results of the entropy measurement will be different [33]. Furthermore, agreement concerning the clinical significance of HRV nonlinear analysis has yet to be reached [34,35,36,37]. Therefore, before the clinical importance of nonlinear HRV parameters is widely accepted, it is of little significance to use them to analyze the consistency of the results. Table 2 lists and defines the parameters used in this study.

To quantify the agreement between the HRV variables extracted from the intervals between successive beats in the BCG, ECG and PW signals, we applied several measures. First, the relative variability between the HRV measurements extracted from the intervals was compared by calculating the differences between the coefficient of variation (CV), calculated from the means and standard deviations (SD) of each pair of HRV features. We defined the quality of the agreement as follows [22]: the absolute value of CV difference < 2%, excellent agreement; 5% > the absolute value of CV difference ≥ 2%, substantial agreement and the absolute value of CV difference ≥ 5%, poor agreement.

Lin’s Concordance Correlation Coefficient (LCCC) was used to evaluate the agreement between each pair of readings by measuring the perpendicular distance between each pair and the line of identity that would result if the two methods agreed perfectly [38,39]. In contrast to the figures given in Reference [40], we applied a more conservative approach [12], and defined the quality of the agreement as follows: excellent agreement if LCCC > 0.99, substantial agreement if 0.99 ≥ LCCC > 0.95; and poor agreement if LCCC ≤ 0.95. As a graphical representation of the concordance between each of the test methods and the gold standard (ECG), we plotted all pairs of heart intervals against the 45° line.

Bland-Altman analysis is often used to evaluate the concordance between two sets of quantitative data obtained by different methods [41]. In this case, we took the ECG as the gold standard and compared each of the test methods (BCG and PW) separately against the ECG by plotting the mean of each pair of readings against the difference between them. As usual, we defined the limits of agreement (LoA) as the 95% confidence intervals of the overall mean of the differences between each pair of readings. The Bland-Altman ratio (BA ratio) was calculated by dividing half the range of the LoA by the overall mean, as defined above. The standard for evaluating the agreement between two methods based on BA ratio value, as given in References [42,43], is: BA ratio < 0.1, excellent agreement; 0.2 > BA ratio ≥ 0.1, moderate agreement and BA ratio ≥ 0.2, poor agreement.

If the quality of agreement given by the above methods differed, we took the worst as the result, and only considered a result to be good if all methods showed a good agreement.

## 3. Results

### 3.1. Consistency of Beat-to-Beat Interval

Consistency, in this context, means the degree to which the statistical characteristics of the heart intervals obtained by different methods agree. These characteristics are shown in Table 3 for all 50 subjects measured over the short-term. Note the similarity of the coefficients of variance of all three modalities in each case. Figure 3 is a graphical depiction of the short-term, beat-to-beat intervals for all 50 subjects, showing the regression line with its equation and corresponding RMS error, for all recorded heartbeats. The left panel is a plot of the J-J intervals (BCG) against the corresponding R-R (ECG) intervals and the right panel compares the P-P (PW) intervals with the ECG in the same way.

Consistency of the beat-to-beat intervals detected from the three types of signals can be seen from Table 3. In addition, for the 50 subjects, the LCCC of the J-J intervals and the R-R intervals was 0.990, while the BA ratio was 0.020. On the other hand, the LCCC of the P-P intervals and the R-R intervals was 0.993, while the BA ratio was 0.019. Taken together, the results show that there is a good agreement between the beat-to-beat intervals.

### 3.2. Consistency of the Short-Term Data

In the first part of the experiment, we calculated the HRV parameters of 50 subjects from the R-R, J-J, and P-P intervals and analyzed the agreement among them. Table 4 contains the mean differences (±their associated SD) between the CV, LCCC, LoA from the Bland-Altman analysis (see Appendix A, Figure A1 and Figure A2), and the BA ratio. We found that the HRV parameters mean, SDNN and LF provided an excellent agreement between the ECG and BCG results, the pNN50 indicated substantial agreement, and the HF, LF/HF and RMSSD showed poor agreement. Table 5 lists the corresponding HRV parameters derived from the P-P and R-R intervals. The parameters mean, SDNN and LF showed the same excellent agreement between the PW and ECG intervals. Although the absolute value of CV difference and BA ratio reflected its good consistency, the value of LCCC is slightly less than 0.99. The other parameters have a larger absolute value of CV difference and this is confirmed by the lower LCCC values for pNN50, RMSSD, HF and LF/HF, which nevertheless all show a substantial agreement. The BA ratio for RMSSD and LF/HF show a moderate agreement, while the parameters pNN50 and HF agree poorly. Figure 4 shows the pairs of HRV metrics for the BCG (J-J) intervals and ECG (R-R) intervals obtained from the 50 subjects plotted against the line of identity. Figure 5 is a similar plot, in which the PW (P-P) intervals metrics are compared with their corresponding ECG values.

### 3.3. Consistency of the Medium-Term Data

Table 6 shows the agreement of HRV parameters calculated from J-J intervals and R-R intervals, for the medium-term data. The results are similar to those of the short-term measurements, with parameters such as Mean, SDNN and LF in good agreement. However, the agreement of both the LCCC and BA ratio value for the parameters pNN50 and LF/HF was only moderate. The BA ratio of RMSSD also indicated moderate agreement while, for the LCCC value, the agreement was poor. For the parameter HF, LCCC value and BA ratio both indicated a moderate agreement. Table 7 summarizes the agreement between the HRV parameters calculated from the P-P and R-R intervals for subject A’s medium-term data. The parameters mean, SDNN and LF agree well, as shown by the LCCC and BA values. The absolute value of CV difference for the parameters pNN50 are not small; the LCCC values for pNN50 suggest a moderate agreement. The LCCC of RMSSD shows a poor agreement, while the BA ratio indicates moderate agreement. For parameters such as HF and LF/HF, the LCCC values show a moderate agreement, while the BA ratio indicates a good agreement. The data for the other two subjects (B and C) who underwent medium-term measurements are tabulated in Appendix B, and the Bland-Altman plots are presented in Figure A3, Figure A4, Figure A5, Figure A6, Figure A7 and Figure A8 of Appendix A. The pairs of HRV metrics obtained from the three subjects’ medium-term BCG and ECG data are shown graphically in Figure 6, with the results from each of the three subjects shown in different colors. Figure 7 is a similar plot, showing the relationship between the PW and ECG data.

## 4. Discussion

The need for simple, low-cost and non-invasive methods to monitor the cardiovascular system has encouraged the development of PW systems as an alternative to ECG [14,15,44]. In recent years, this need has also led to the re-emergence of BCG acquisition systems [45,46,47,48]. However, ECG and PW systems currently require that the sensors are either applied manually to the patient’s skin or, for longer-term use, are held in position with an adhesive patch, a strap or a more elaborate mechanical device such as the one used in this study for PW measurements. In contrast, BCG sensors can be fully integrated into everyday objects such as beds, chairs or weighing scales [49]. Furthermore, BCG signals can provide additional information about the subject, such as sleeping position, gross movement and tremor [50].

Although BCG has been investigated as a potential surrogate for ECG in HRV analysis [10,11,12,13,18], it is not yet widely accepted and there is currently no agreement about how best to quantify differences between the two techniques. Correlation coefficient, BA analysis or relative error is commonly used for assessment of the association between the heart intervals and the calculation of HRV. However, in some cases, the results may be well correlated, but do not necessarily closely agree. Some studies that have explored the agreement between ECG and BCG signals for calculating HRV parameters have only analyzed the heart intervals [51]. Thus, a further detailed analysis of derived HRV parameters is still required. This study is an attempt to more comprehensively explore the agreement between the HRV parameters calculated from the J-J and R-R intervals.

Analyzing the short-term data from 50 subjects and the longer-term data from three, we can conclude that the parameters such as mean, SDNN, and LF, used to describe the agreement between HRV measured by BCG and ECG, are closely consistent with each other; pNN50 showed moderate consistency, whereas RMSSD, HF and LF/HF agree poorly. Shin et al. [13] think that the errors in HF, LF/HF and pNN50, obtained from BCG and ECG signals, are relatively high, a similar finding to ours, except for pNN50. The reason for this slight difference is that both data analysis methods and the values of the boundaries between good, moderate and poor agreement used in the Shin study are not the same as those that we adopted. When comparing the PW and ECG results (i.e., P-P and R-R intervals), the consistency of the mean, SDNN and LF were good, while the consistency of pNN50, RMSSD, HF and LF/HF was poor. The SDNN value reflects the overall activity of the autonomic nervous system. The larger the value, the higher the overall activity and the stronger its regulatory ability. In the results presented here, the values of SDNN derived from the J-J, R-R and P-P intervals are highly consistent, which indicates that BCG and PW may perform as well as ECG in evaluating the risk of heart disease. LF is an indicator of the overall regulatory ability of the sympathetic and parasympathetic nervous system and is often used to analyze the fatigue state.

Overall, the agreement between the J-J and P-P intervals in calculating HRV parameters, was close, with both giving consistent results with the ECG measurements. For pNN50, the consistency of the J-J intervals was good, whereas the consistency of the P-P interval was poor. It is notable that the parameters RMSSD and HF, calculated from the J-J intervals do not agree well with those calculated from R-R intervals, a similar result to other reported findings [12,52]. This is caused by the low-frequency components of the BCG signals, which introduce inaccuracies when detecting the blunter peaks of the BCG J waves compared to the more clearly defined peaks in the ECG R-wave [53].

A limitation to this study is the detection of spurious maxima in the BCG signal. Table 1 lists the number of false peaks detected by each method. The errors in the BCG signals occur more frequently because it is difficult for subjects to remain still, especially during the long-term acquisition of signals. Body movement will affect the BCG signal, cause the waveform to be distorted, and render the peak detection algorithm ineffective. In general, BCG based measurements are more widely used for sleep monitoring, office health monitoring and other routine heart rate measurements. If the body motion artefacts in the BCG signals can be effectively identified and filtered, the routine monitoring of BCG signals will be more widely used. This is the focus of our future work. For instance, a DnCNN network can be used to construct a BCG signal denoising model, and residual mapping can be added to enable the neural network to learn the characteristics of the noise [54]. Another limitation of the study is that the participants were all healthy young adults. In future work, we will analyze more subjects over a greater range of ages and, in particular, we will carry out daily monitoring of patients with heart disease.

## 5. Conclusions

In this study, we have performed experiments to explore the consistency of HRV analysis based on BCG signals to investigate the possibility of using this approach to replace or complement ECG and PW HRV measurements. We employed a wider variety of statistical tests such as CV, LCCC and Bland-Altman methods than in other studies and also performed measurements on more subjects than those of other studies and used strict comprehensive evaluation criteria, which improved the reliability. The results show the HRV parameters mean, SDNN and LF, obtained from BCG signal J-J intervals, agree well with the R-R intervals from simultaneous ECG measurements. The parameters pNN50 agree only moderately well, while the RMSSD, HF and LF/HF agree poorly. When compared with HRV analysis based on PW signals, BCG had a better performance, as measured by (pNN50). Coupled with the fact that the BCG signal acquisition device has less impact on the user, this technique shows promise for routine heartrate monitoring. In addition, we performed measurements on more subjects than other studies and used strict comprehensive evaluation criteria, which improved the reliability and practicability of the work described in this paper.

## Figures and Tables

**Figure 1 sensors-22-02423-f001:**
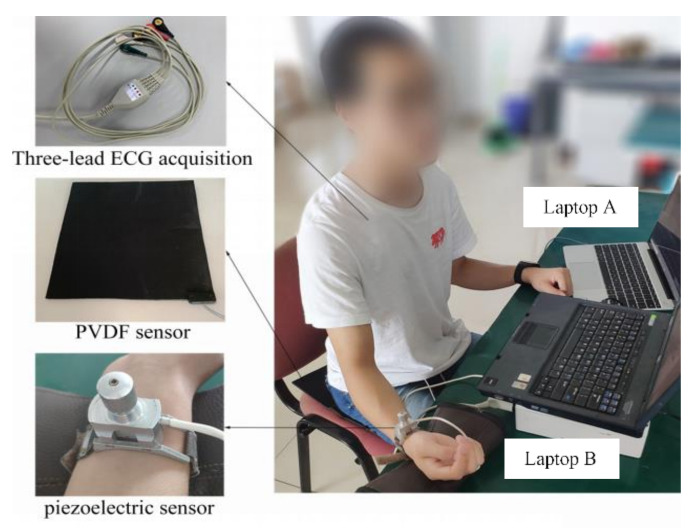
Photograph of the devices used to simultaneously record BCG, ECG and PW signals.

**Figure 2 sensors-22-02423-f002:**
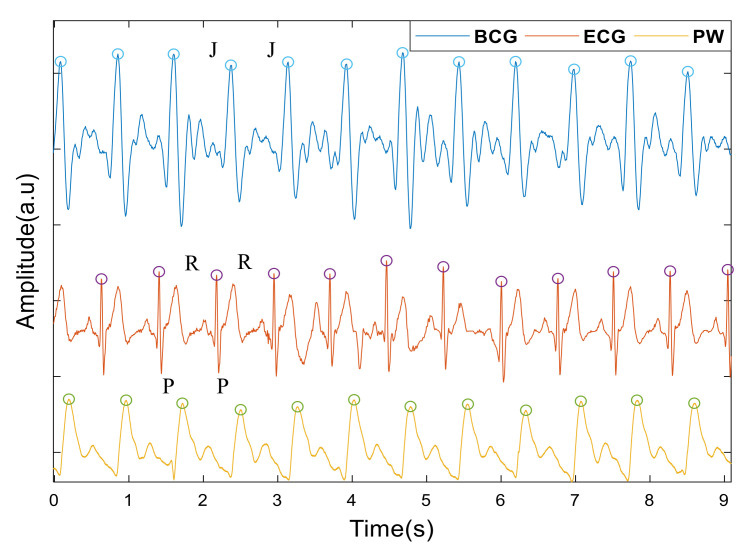
Peak detection of the BCG, ECG, PW signals. Circles indicate time points from which heart intervals are recorded. J for the BCG signals, R for the ECG and P for pulse wave (PW).

**Figure 3 sensors-22-02423-f003:**
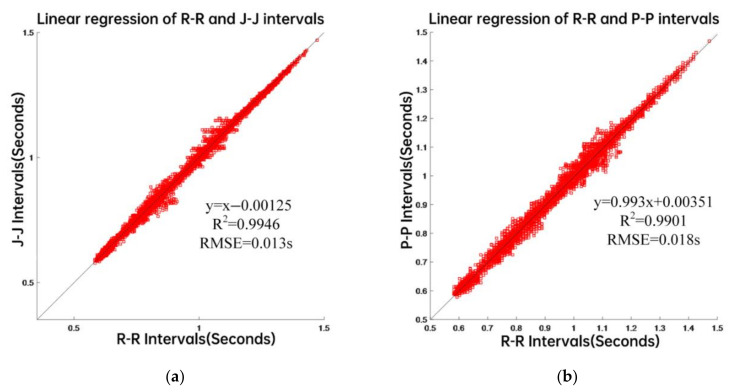
Data from 50 subjects. (**a**) linear regression of R-R and J-J intervals. (**b**) linear regression of R-R and P-P intervals.

**Figure 4 sensors-22-02423-f004:**
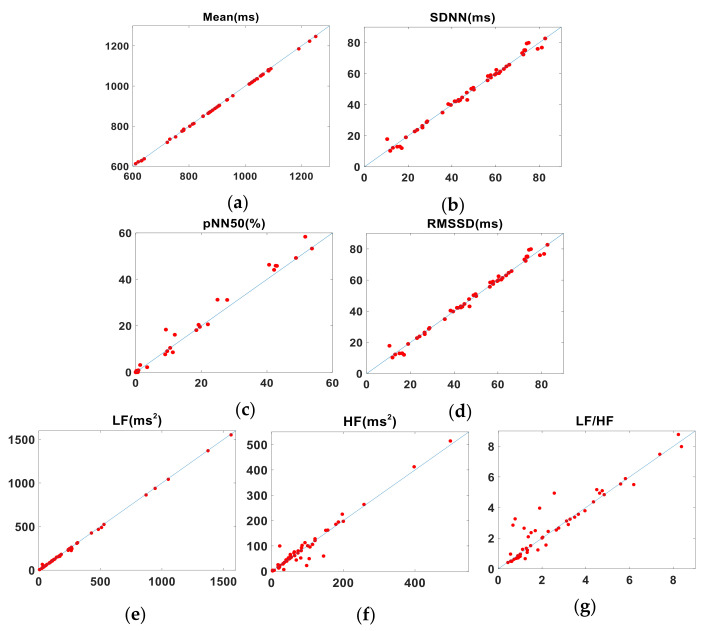
Pairs of HRV metrics (title above each plot) obtained from 50 subjects plotted against the line of identity (representing perfect agreement). *x*- and *y*-axes show BCG and ECG values, respectively. (**a**) Mean. (**b**) SDNN. (**c**) pNN50. (**d**) RMSSD. (**e**) LF. (**f**) HF. (**g**) LF/HF.

**Figure 5 sensors-22-02423-f005:**
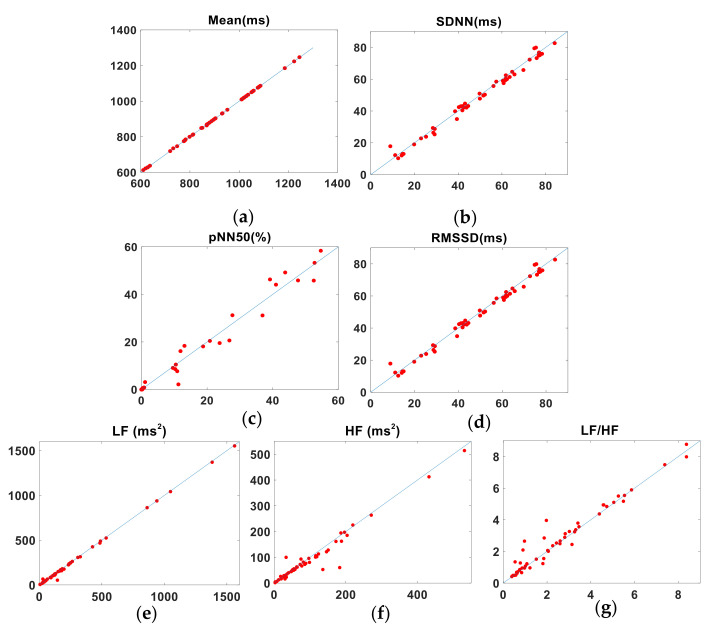
Pairs of HRV metrics obtained from 50 subjects plotted against the line of identity (representing perfect agreement). *x*- and *y*-axes show PW and ECG values, respectively. (**a**) Mean. (**b**) SDNN. (**c**) pNN50. (**d**) RMSSD. (**e**) LF. (**f**) HF. (**g**) LF/HF.

**Figure 6 sensors-22-02423-f006:**
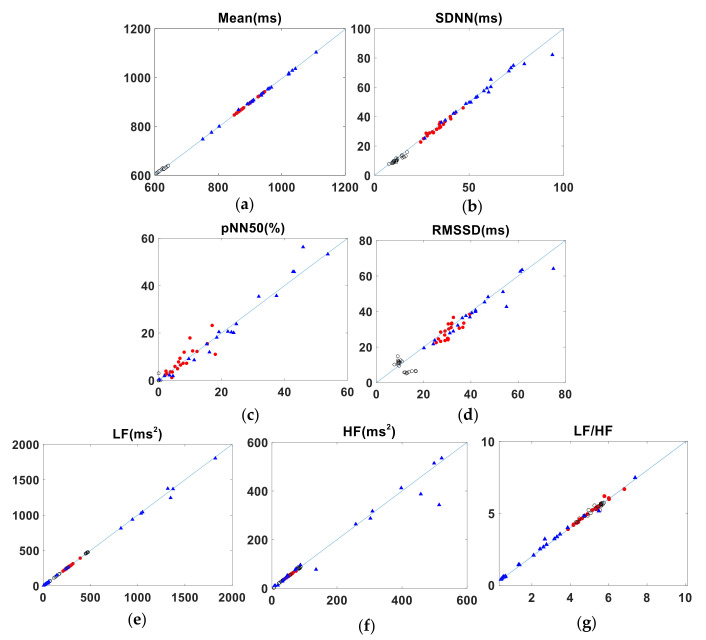
Pairs of HRV metrics obtained from the medium-term data plotted against the line of identity. *x*- and *y*-axes show BCG and ECG values, respectively. (Results from each of the three subjects are shown in different colors.). (**a**) Mean. (**b**) SDNN. (**c**) pNN50. (**d**) RMSSD. (**e**) LF. (**f**) HF. (**g**) LF/HF.

**Figure 7 sensors-22-02423-f007:**
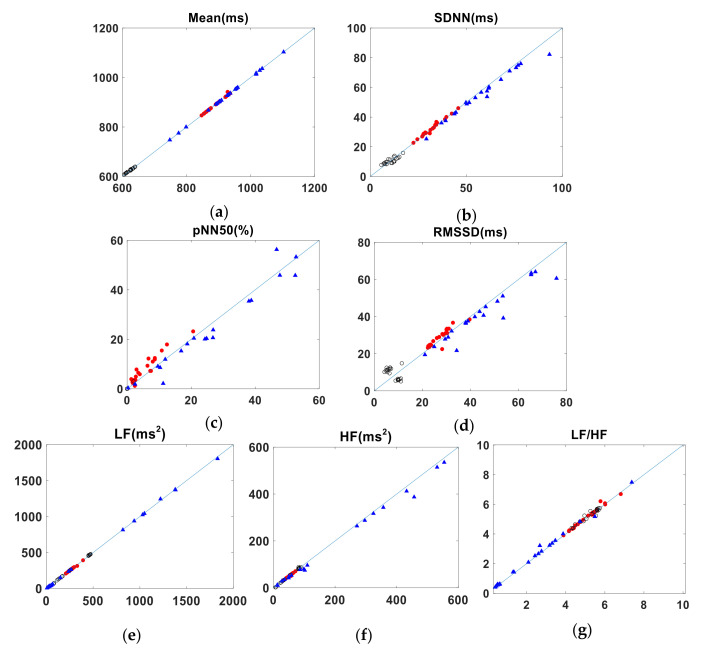
Pairs of HRV metrics obtained from the medium-term data plotted against the line of identity. *x*- and *y*-axes show PW and ECG values, respectively. (Results from each of the three subjects are shown in different colors.) (**a**) is for Mean. (**b**) is for SDNN. (**c**) is for pNN50. (**d**) is for RMSSD. (**e**) is for LF. (**f**) is for HF. (**g**) is for LF/HF.

**Table 1 sensors-22-02423-t001:** Number of spurious peaks in the data.

	Total Interval Number	Number of Spurious Peaks		
		ECG	PW	BCG
50 subjects	17,045	31	22	78
Subject A	6015	21	14	197
Subject B	6593	13	18	93
Subject C	10,714	9	8	107

**Table 2 sensors-22-02423-t002:** Definition of the HRV parameters.

	Parameters	Definition
Time Domain	Mean (ms)	Average of heart intervals
	SDNN (ms)	Standard deviation of heart intervals
	pNN50 (%)	Percentage of the difference between adjacent heart intervals that exceed 50 ms
	RMSSD (ms)	Root mean square of the difference between adjacent heart intervals
Frequency Domain	LF (ms^2^)	Spectral power in the low-frequency range (0.04~0.15 Hz)
	HF (ms^2^)	Spectral power in the high-frequency range (0.15~0.40 Hz)
	LF/HF	Ratio of LF to HF

**Table 3 sensors-22-02423-t003:** Descriptive statistics of the intervals obtained from the short-term data recordings of 50 subjects.

	N	Min (ms)	Max (ms)	Mean (ms)	SD (ms)	CV (%)
BCG	17,045	550	1472	883.3	181	20.5
PW	17,045	551	1472	886.3	182.2	20.6
ECG	17,045	550	1469	883.4	181.8	20.6

**Table 4 sensors-22-02423-t004:** Statistical analysis for agreement between HRV parameters calculated from J-J intervals and R-R intervals of 50 subjects.

	BCG	ECG	CV_b_–CV_e_	LCCC	Lower, Upper	BA_Ratio
Mean	917.3 ± 160.3	913.9 ± 158.8	0.11%	**0.999**	−1.544, 8.324	0.003
SDNN	48.8 ± 21.4	49.0 ± 22.0	−1.04%	**0.994**	−4.688, 4.318	0.046
pNN50	10.5 ± 16.3	11.4 ± 17.4	0.64%	**0.990**	−5.115, 3.556	0.169
RMSSD	44.5 ± 26.1	44.4 ± 28.7	−5.32	0.960	−15.434, 14.867	0.198
LF	254.0 ± 341.9	250.1 ± 339.6	−1.16%	**0.999**	−14.261, 22.111	0.036
HF	93.7 ± 93.2	92.6 ± 97.7	−6.01%	**0.995**	−45.242, 47.466	0.249
LF/HF	2.5 ± 2.0	2.8 ± 2.0	5.4%	0.934	−1.639, 1.218	0.265

CV_b_: the coefficients of variation of BCG; CV_e_: the coefficients of variation of ECG; Lower, Upper: The range of agreement limits in the Bland-Altman analysis. Bold figures indicate excellent agreement, as defined in the text.

**Table 5 sensors-22-02423-t005:** Statistical analysis for agreement between HRV parameters calculated from P-P intervals and R-R intervals of 50 subjects.

	PW	ECG	CV_p_–CV_e_	LCCC	Lower, Upper	BA_Ratio
Mean	913.7 ± 159.0	913.9 ± 158.8	0.03%	**1**	−2.484, 2.014	0.001
SDNN	49.6 ± 22.0	49.0 ± 22.0	−0.52%	**0.993**	−4.241, 5.435	0.049
pNN50	11.4 ± 17.1	11.4 ± 17.4	−4.09%	0.986	−5.490, 5.710	0.246
RMSSD	44.4 ± 26.4	44.4 ± 28.7	−4.59	0.98	−11.143, 10.335	0.12
LF	255.8 ± 340.3	250.1 ± 339.6	−1.76%	**0.999**	−23.131, 34.590	0.057
HF	100.8 ± 102.2	92.6 ± 97.7	−4.06%	0.968	−39.132, 55.485	0.245
LF/HF	2.6 ± 2.1	2.8 ± 2.0	4.93%	0.972	−1.084, 0.772	0.17

CV_p_: the coefficients of variation of BCG; CV_e_: the coefficients of variation of ECG; Lower, Upper: The range of agreement limits in the Bland-Altman analysis. Bold figures indicate excellent agree ment, as defined in the text.

**Table 6 sensors-22-02423-t006:** Statistical analysis for agreement between HRV parameters calculated from subject A’s J-J intervals and R-R intervals.

	BCG	ECG	CV_b_–CV_e_	LCCC	Lower, Upper	BA_Ratio
Mean	931.8 ± 90.2	927.4 ± 88.9	0.1%	**0.998**	−1.243, 9.963	0.003
SDNN	59.6 ± 18.5	55.79 ± 15.0	0.43%	**0.998**	−5.186, 6.793	0.053
pNN50	22.4 ± 15.7	22.3 ± 17.6	−3.53%	0.981	−6.339, 6.474	0.143
RMSSD	44.3 ± 16.3	41.1 ± 13.9	3.01%	0.916	−7.450, 13.763	0.124
LF	534.9 ± 604.9	528.5 ± 599.3	−0.3%	**0.999**	−47.409, 60.029	0.051
HF	193.5 ± 192.4	181.5 ± 180.5	−0.02%	0.971	−73.307, 97.324	0.228
LF/HF	2.5 ± 1.8	2.6 ± 1.9	−0.28%	0.974	−0.901, 0.692	0.159

CV_b_: the coefficients of variation of BCG; CV_e_: the coefficients of variation of ECG; Lower, Upper: The range of agreement limits in the Bland-Altman analysis. Bold figures indicate excellent agreement, as defined in the text.

**Table 7 sensors-22-02423-t007:** Statistical analysis of the agreement between HRV parameters calculated from subject A’s P-P interval and R-R interval.

	PW	ECG	CV_p_–CV_e_	LCCC	Lower, Upper	BA_Ratio
Mean	927.6 ± 89.2	927.4 ± 88.9	**0.03**%	**0.999**	−1.929, 2.344	**0.001**
SDNN	58.4 ± 16.0	55.79 ± 15.0	**0.56**%	**0.994**	−2.245, 7.425	**0.042**
pNN50	24.2 ± 16.9	22.3 ± 17.6	−8.75%	0.971	−5.420, 9.153	0.157
RMSSD	44.9 ± 15.0	41.1 ± 13.9	−0.2%	0.918	−5.258, 12.769	0.105
LF	532.4 ± 601.1	528.5 ± 599.3	**−0.5**%	**0.999**	−13.490, 21.298	**0.016**
HF	194.0 ± 188.7	181.5 ± 180.5	−2.17%	0.984	−17.919, 42.956	0.081
LF/HF	2.5 ± 1.9	2.6 ± 1.9	2.26%	0.986	−0.358, 0.216	0.057

CV_p_: the coefficients of variation of BCG; CV_e_: the coefficients of variation of ECG; Lower, Upper: The range of agreement limits in the Bland-Altman analysis. Bold figures indicate excellent agreement, as defined in the text.

## Data Availability

The raw data supporting the conclusions of this article will be made available by the corresponding author upon reasonable request.

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
