# Peer review of "Statistical Analysis of the Consistency of HRV Analysis Using BCG or Pulse Wave Signals"

_sensors, 2022, doi:10.3390/s22062423_

Round 1
Reviewer 1 Report
The authors have performed experiments to explore the consistency of HRV analysis based on BCG signals to investigate the possibility of using this approach by showing the HRV parameters obtained from ECG, BCG, and PW signals. The reported results seemed promising, but there are several critical concerns that require at least a major revision.
- Lines 99-101: You need to explain more about the reason why Pearson correlation and the relative error cannot comprehensively describe either the consistency or the lack of agreement between the two methods. Add relevant references.
- Line 125-127: Ethical concerns about this study is not described in this paper. For example, if you followed the IRB, please add the IRB number.
- For lines 114-116, they look like normal sentences. Please change the format so readers can recognize that they are two main contributions of the paper.
- In lines 134-142, add references or spscifications for all the devices, e.g., BOBO Technology, device for the synchronous recording, laptop A, and laptop B.
- line 348: How did you select 3 subjects: subjects A, B, and C? Are they representative subjects of 50 subjects? Can you add explanation on this?
- line 345: How do you define spurious maxima and how do you detect it?
- line 26: 32 females + 28 males become 60 subjects not 50 subjects? Which is the correct one?
- line 31-32: good agreement and poor agreement are not quantitative terms. Can you describe these using quantitative terms?
Reviewer 2 Report
The present paper analyzes the agreement between inter-beat-intervals (IBIs) computed from electrocardiography (ECG), ballistocardiography (BCG), and pulse-wave signal (PW, or photoplethysmography, PPG). In particular, the authors considered the R-R intervals from the ECG to be the gold standard. Thus, they measured the agreement between the R-R intervals and the J-J intervals of the BCG, as well as the concordance between the former and the IBIs computed from PPG (P-P intervals). Although the paper is scientifically sound (with some exceptions documented later), several parts must be revised carefully to make the results more complete and reliable.
- In the last sentence of the abstract, it is not clear if the statement "they have almost the same performance except pNN50" refers to the comparison between ECG and PW, ECG and BCG, or both. This sentence appears incorrect in all cases since both BCG and PW have demonstrated poor agreement with several HRV parameters other than pNN50, such as HF (Section 3).
- In the Introduction, the authors stated that previous studies on the agreement between ECG and BCG measures of HRV neglected non-linear parameters. Therefore, why did they limit their attention to SD1 and SD2 indices? Since time windows of 5 minutes are generally deemed enough to evaluate many other HRV parameters of this kind (Castaldo 2019), they could include, for example, several others among approximate entropy, sample entropy, detrended fluctuation analysis, and correlation dimension. This will certainly improve the significance of their results.
- In Section 2, while the BCG system is well described (p.3, lines 134-138), no details are provided for the devices used for ECG and PW recording; even the manufacturer is omitted. Moreover, no specifications on the available bandwidth are provided for the three signals. The bandwidth information is essential to correctly interpret the differences detected between signals and identify potential limitations of the study.
- The approach used to evaluate "long-term HRV" does not convince me entirely. When assessing long-term variability, the HRV parameters of interest (typically, frequency-domain and non-linear ones) are calculated from a single HRV segment of long duration, usually 24-hours (Malik 1996, Shaffer 2017). Assuming 100 minutes to be enough to evaluate long-term HRV changes, in this paper, "long-term HRV parameters" are computed from 20 frames of 5 minutes each for every subject (p.4 lines 154-156). Therefore, what the authors actually did was just compute 20 samples of short-term HRV parameters over time. This is very similar to the analysis they reported in Section 3.1, with the exception that, this time, they considered only 20 repeated measures from each of the three subjects instead of 50 samples drawn from the population under study. In summary, I don't think this is a proper way to assess "long-term" HRV, as the authors claim they have done (p. 3, line 132).
- Page 5, lines 177-179. The percentage of beats corrected in this manner should be declared.
- I could not find any mention of the adopted BAratio thresholds (p. 6, lines 211-212) in reference [27]. Either the citation is erroneous, or I overlooked it. Could the authors provide the exact quote for these thresholds? Moreover, could they illustrate the rationale behind the BAratio metric or add adequate in-text referencing for it? Regarding the coefficient of variation (CV), this coefficient is not even named in [12], so I suppose this could be another erroneous reference. The authors should double-check the references for all the thresholds presented in Section 2. Those thresholds that do not have any relevant reference should be motivated. Finally, I could not find the manuscript referenced in [25] on the internet. Perhaps, when defining the thresholds for the LCCC, the authors should also cite Akoglu 2018, who points to the same thresholds and whose paper can be easily retrieved by interested readers.
- In [27], a method to account for repeated measures in Bland-Altman (BA) analysis is illustrated. Other approaches to deal with repeated measures designs have been suggested in the literature (e.g., Bland Altman 1999). Did the authors consider any of such repeated measures adjustments when calculating Limits of Agreement and BAratio?
- There is no need to report both SD1 and RMSSD indices in the Results as they are totally equivalent measures (see, e.g., Ciccone 2017). In particular, these two indices should be perfectly linearly correlated (i.e., Pearson's rho = 1), which makes me wonder why the LCCC of RMSSD looks so different from that of SD1 (0.960 vs. 0.902 in Table 3). It could be intrinsic of how the LCCC is calculated but, please, double-check the results of both indices. Besides, there is a mistake in Table 3: the title in the first column should be "BCG" instead of "PW" (other tables in the Appendices show similar typos).
- Among the results, the authors wrote (p.7, lines 243-244): "The parameters Mean, SDNN, LF and SD2 show the same excellent agreement between the PW and ECG intervals". Actually, given its LCCC value, SD2 should be listed among the parameters with "substantial" agreement.
- From the Results (p. 9, lines 275-278): "Although the CV differences for the parameters pNN50 and SD1 are not small, the LCCC values for pNN50 and RMSSD suggest a moderate agreement. The LCCC of RMSSD shows a poor agreement...". There is something wrong with this sentence. There are other erroneous statements like this all around the manuscript. Check the entire paper carefully.
Reviewer 3 Report
The manuscript proposes an approach based on comprehensive statistical analysis, including coefficients of variation, Bland-Altman analysis, and Lin’s concordance correlation coefficient, and 28 were utilised to analyse the consistency of BCG and ECG signals in HRV analysis. The proposed work has presented a significant idea of alternative to HRV analysis. However, there are some suggestions which need to be incorporated in the manuscript for better presentation of the work.
- Author needs mention the final numeric results in the abstract section.
- What is pNN50? Please incorporate in the abstract section.
- Use of “recruited” word in the abstract needs to replace with more appropriate word.
- The use of “cardiovascular disease” in entire manuscript should be replaced with “cardiovascular diseases”. Sentence like “It has been known for at least 50 years that changes in HRV are strongly associated with cardiovascular disease such as heart failure, coronary artery disease, acute myocardial infarction and essential hypertension and more recent studies have confirmed this”.
- Author is suggested to used uniform citation style in the entire manuscript. Need to correct in “Alba et al. explored the feasibility of HRV analysis using 82 the J-J, I-I, K-K, and H-H intervals of BCG signals” and also in many more.
- What is the significance of extracting time domain, frequency domain, and non-linear analysis parameters in the proposed study.
- Author uses the citations in the conclusion section which is not a scientific way of writing.
- Author needs to compare the achieved results with existing studies
- Author suggested to incorporate the highlights and limitations of the proposed work.
- The entire manuscript should be refined for English grammatical structure and phraseology.
Round 2
Reviewer 2 Report
The authors addressed several of my concerns, and the paper has improved significantly compared to the original. Still, I believe some points remain to be considered for the manuscript to be accepted.
Response to Concern #1: In response to my first comment, the authors revised the final lines of the abstract. However, the last sentence still appears unclear. I suggest rephrasing it as follows: "Additionally, the R-R intervals were compared with P-P intervals extracted from the pulse wave (PW). Except for pNN50, which exhibited poor agreement in this comparison, performances of the HRV indices estimated from the PW and the BCG signals were similar."
Response to Concern #2: Responding to my second comment, the authors removed the evaluation of SD1 and SD2 and explained why they did not include any other non-linear HRV index in the analyses.
I find the choice to neglect non-linear parameters hard to understand as, in the Introduction, the authors themselves highlight the lack of non-linear HRV analyses as one of the main limitations in the study by Brueser et al. [12] (lines 93-95 on the revised manuscript). Moreover, their justification for including only SD1 and SD2 in the original version of the manuscript (i.e., because these non-linear indices "do not need the data to be stationary") is not convincing. In fact, even frequency-domain HRV features, such as LF, HF, and LF/HF, assume second-order stationarity of the HRV signal (see, e.g., Magagnin 2011) and require specific HRV processing to be calculated. Nevertheless, they have been considered in the study. Consequently, I strongly suggest including some non-linear HRV parameters in the analysis, as requested, or providing more objective (i.e., protocol-related) reasons for not doing so. In the latter case, the authors will describe, in the paper, such objective reasons and include this choice as an explicit limitation of their study.
Response to Concern #3: I appreciate the additional details the authors provided. However, the sampling rate of a signal and its passband frequencies (i.e., the so-called "signal bandwidth") are two different things. Please, add in the paper the second information (passband frequencies) for all the signals collected during the experiment.
Lastly, Bland-Altman (BA) plots referring to the same HRV index in Appendix A should be provided with the same Y-axis range (or "limits"). For example, the BA plot of the "Mean (ms)" feature should be limited between ymin=-3 and ymax=10, both in figures A1 and A2. The same should be done for corresponding HRV features in the other figure pairs (A3,A4; A5,A6; A7,A8). This solution will facilitate comparisons between BA plots of the BCG-ECG analysis and those related to the PW-ECG one.
References:
Magagnin V, Bassani T, Bari V, Turiel M, Maestri R, Pinna G D and Porta A 2011. Non-stationarities significantly distort short-term spectral, symbolic and entropy heart rate variability indices. Physiol. Meas. 32 1775–86
Reviewer 3 Report
Authors have included all suggestions
Author Response
Thank you very much for your comments.